# The Role and Appropriate Selection of Guidewires in Biliopancreatic Endoscopy

**DOI:** 10.3390/medicina61050913

**Published:** 2025-05-18

**Authors:** Daniele Alfieri, Claudia Delogu, Stefano Mazza, Aurelio Mauro, Erica Bartolotta, Alessandro Cappellini, Davide Scalvini, Francesca Torello Viera, Marco Bardone, Andrea Anderloni

**Affiliations:** 1Gastroenterology and Endoscopy Unit, IRCCS Foundation Policlinico San Matteo, 27100 Pavia, Italy; daniele.alfieri01@universitadipavia.it (D.A.); claudia.delogu01@universitadipavia.it (C.D.); aurelio.mauro@smatteo.pv.it (A.M.); erica.bartolotta01@universitadipavia.it (E.B.); alessandro.cappellini01@universitadipavia.it (A.C.); davide.scalvini01@universitadipavia.it (D.S.); f.torello@smatteo.pv.it (F.T.V.); m.bardone@smatteo.pv.it (M.B.); a.anderloni@smatteo.pv.it (A.A.); 2Department of Internal Medicine and Therapeutics, University of Pavia, 27100 Pavia, Italy

**Keywords:** ERCP, EUS, biliopancreatic endoscopy, biliary drainage, difficult biliary cannulation, hepaticogastrostomy, biliary stricture, pancreatic cannulation

## Abstract

Guidewires are indispensable tools in biliopancreatic endoscopy, playing a critical role in facilitating access and enabling the advancement of various devices during interventions such as Endoscopic Retrograde Cholangiopancreatography (ERCP) and Endoscopic Ultrasound (EUS)-guided procedures. These devices are primarily used to achieve and maintain access to lumens, ensuring the success of complex therapeutic maneuvers. Guidewires vary widely in terms of material, structure, length, diameter, and tip shape, offering distinct advantages depending on the clinical context. Therefore, selecting the appropriate guidewire is crucial and must be tailored to the specific requirements of each procedure. This article provides a comprehensive review of the current landscape of guidewire use in biliopancreatic endoscopy, emphasizing their importance, characteristics, and best practices for selection to optimize patient outcomes. By reviewing existing guidelines and the literature, this paper aims to enhance the endoscopist’s understanding of guidewire technology and its application in biliopancreatic endoscopy.

## 1. Introduction

Biliopancreatic endoscopy plays a pivotal role in the diagnosis and treatment of a broad spectrum of disorders. Among the various tools employed in this field, guidewires are indispensable for facilitating access to key anatomical structures and enabling the precise advancement of catheters, stents, and other medical devices [1]. Their importance is particularly evident in procedures such as Endoscopic Retrograde Cholangiopancreatography (ERCP) and Endoscopic Ultrasound (EUS)-guided interventions, where the design, quality, and performance of the guidewire significantly impact the safety, efficiency, and overall success of the procedure. Guidewires differ in material composition, length, diameter, tip shape, and stiffness of both the tip and shaft, allowing for optimal selection based on procedural requirements (Table 1). A wide range of guidewires is available in the market, from those designed for general use to highly specialized variants tailored for specific indications (Table 2). The effectiveness of ERCP and EUS-guided interventions is often closely tied to the proper choice and skillful use of guidewires. Given their critical role in both diagnostic and therapeutic outcomes, a thorough understanding of guidewire characteristics, classifications, and best practices is essential for endoscopists. This review examines the recent literature focusing on the functions and attributes of guidewires in ERCP and EUS-guided procedures. Additionally, it discusses potential complications and the latest advancements aimed at stabilizing guidewires and minimizing procedure-related risks.

## 2. Guidewire Characteristics

### 2.1. Material Composition and Structure

Guidewires used in gastrointestinal applications are typically classified into three types: monofilament, spiral, and coated or sheathed [2]. Monofilament wires are usually made of stainless steel for rigidity, whereas spiral wires combine a monofilament core and an outer spiral coil for flexibility and durability. Coated or sheathed wires, the most used in the biliopancreatic endoscopy, feature a monofilament core and an outer smooth polymer coating [2]. Most guidewires have a nitinol core, a nickel–titanium alloy offering shape memory for greater flexibility and shape recovery, essential for navigating tortuous ducts. The outer coating typically includes hydrophobic materials such as polytetrafluoroethylene (PTFE) or polyurethane, which reduce friction to facilitate seamless accessory exchanges, offer electrical insulation, and enhance fluoroscopic visualization [3]. Guidewires consist of a rigid shaft and a 5–10 cm long tip whose core is made of flexible, radiopaque materials like platinum or tungsten. Some guidewires include a spiral-coiled spring at the tip to improve flexibility and traceability [3]. A design that features a more flexible, hydrophilic tip combines the advantages of both material types: a long, rigid portion for supporting accessories and a soft, adaptable tip for navigation through ducts. The hydrophilic coating, often applied to the tip or the entire length, facilitates smooth cannulation, especially in the biliopancreatic tract, by becoming highly lubricious when activated with flushing. Fully hydrophilic guidewires offer better performance in case of difficult papilla or stenosis but require continuous lubrication and are more prone to displacement [4]. A randomized study published in 2008 demonstrated that the use of a fully hydrophilic guidewire as a first-line strategy significantly improved the success rate of selective bile duct cannulation compared to conventional approaches. Notably, after crossover, the overall cannulation success rates equalized between groups. Hydrophilic guidewires are particularly advantageous in navigating tight stenoses, especially in the setting of hilar strictures, where their enhanced trackability facilitates selective access. However, the study was limited by its single-center design, small sample size, lack of blinding, and absence of long-term follow-up to assess durability and complication rates [5]. Lastly, the guidewire body typically has a spiral striping pattern, aiding in endoscopic visualization and coordination during accessory exchanges.

### 2.2. Length

Traditionally, ERCP is performed using the long-wire system using guidewires of 420–480 cm (typically 450 cm), allowing wire extension through the endoscope and external accessory attachment [6]. Advantages include wide compatibility across devices and manufacturers and the assistant-applied countertraction to aid passage in complex cases such as tight strictures. Additionally, some authors suggest that this approach may be better suited for specific circumstances, such as EUS-assisted rendezvous procedures and surgically altered anatomies (e.g., Roux-en-Y gastric bypass or hepaticojejunostomy), in which long wires with a stiff shaft are crucial for maintaining access during prolonged procedures and multiple device exchanges through long scope working channels [1,7]. On the other hand, the long-wire system requires significant coordination between the endoscopist and an experienced assistant; poor communication can prolong procedure time, increasing radiation exposure and the risk of biliary access loss. To overcome these issues, advances in wire-locking mechanisms and catheter technology have enabled the development of short-wire systems [8]. The three most common short-wire systems widely available are the RX System (Boston Scientific, USA) (Figure 1), the Fusion System (Cook Endoscopy, Bloomington, IN, USA), and the V System (Olympus, Japan). These systems enable the locking of a shorter guidewire (260–270 cm in length) both externally at the working channel port and internally at the elevator (except for the RX System). This feature allows the endoscopist to independently control the guidewire at the working channel port, ensuring stable ductal access, reducing procedure time, and minimizing the risk of displacement due to inadvertent countertraction by the assistant [2]. Comparative studies show that physician-controlled short-wire systems are comparable to assistant-controlled long-wire methods in success rate and radiation exposure, with the added benefit of reduced manpower [9,10]. In addition, physician control provides better tactile feedback and real-time adjustments, improving alignment during cannulation, reducing the risk of injury to the ducts, and enabling a quicker detection of complications like pancreatic duct entry or papillary fistula creation. The limitations of this prospective randomized study [9] was that the endoscopist performing ERCPs was aware of the system used and knew that some time measurements were being recorded during the procedures; furthermore all procedures were performed by experts. Regarding safety, Baxbaum et al. demonstrated in a randomized controlled trial (RCT) that endoscopist-controlled cannulation was associated with a significantly lower adverse event rate (2.8% vs. 11.2%; *p* = 0.016), mainly due to a reduction in PEP (2.8% vs. 93%; *p* = 0.049) indipendent of the assistants’ experience, which was notably high (minimum 7 years and median 9 years of ERCP experience). Due to this significative difference in complications, the study was prematurely stopped, leading to a sample size that was less than half of what was originally planned to evaluate the primary efficacy outcome of successful cannulation. Another limitation is that this was a single-center study, which limits the generalizability of the findings [11].

### 2.3. Diameter

The most used guidewire sizes in ERCP are 0.025 and 0.035 inches (approximately 0.6 and 0.9 mm, respectively). The 0.035-inch wire is the standard choice in biliary endoscopy, although the 0.025-inch wire may be preferred for navigating hilar biliary strictures as shown in an RCT by Han et al. comparing different diameter guidewires for selective intrahepatic biliary ducts cannulation. However, the study had several limitations: the high proportion of patients undergoing single-duct drainage, the use of various catheter types, and the inclusion of only 90 out of 170 enrolled patients may have influenced the results [12]. No evidence supports a clear advantage of 0.025-inch guidewires in terms of cannulation success or reducing post-ERCP pancreatitis rates. A meta-analysis comparing the two diameters found similar outcomes, with a relative risk of 1.02 (95% CI, 0.96–1.08) for cannulation success and 1.15 (95% CI, 0.73–1.81) for PEP rates. Additionally, no significant differences were observed for other outcomes, such as the use of techniques like precut sphincterotomy or double guidewire technique, or adverse events such as cholangitis, perforation, or bleeding [13]. Smaller-diameter guidewires, such as 0.018 inches, are sometimes preferred for more delicate procedures, such as Wirsung duct or minor papilla cannulation. However, as the diameter decreases, the wire becomes less rigid, making it more difficult to maneuver and introduce accessories. Once cannulation is successfully achieved with a small-diameter wire, it is typically exchanged with a 0.035-inch guidewire to overcome the limitations of thinner guidewires and facilitate the continuation of the procedure.

### 2.4. Tip Shape

Guidewires for biliopancreatic endoscopy feature with different morphologies of the distal tip: straight, angled, and J-shaped. The length of the tip varies between different versions, typically ranging from 3 to 10 cm. Since the tips are usually hydrophilic, longer tips are believed to offer greater lubrication to facilitate selective cannulation, and radiopaque markers present on the guidewire tip aid in movement and measurements under fluoroscopic guidance. Angled guidewires typically offer high torqueability and tip flexibility, providing selective cannulation, navigation of strictures, and advancement in challenging lumens potentially reducing complications like ductal perforation and PEP, though clinical data are limited [2] (Figure 2). Vihervaara et al. conducted an RCT to compare the efficacy of hydrophilic angled-tip and straight-tip guidewires in biliary cannulation. The study included 155 patients randomized into two groups: 72 with an angled-tipped guidewire and 83 with a straight-tipped guidewire. While the success rate of primary biliary cannulation was similar between the groups (60% vs. 65%, *p* = 0.61), the angled-tipped guidewire achieved deep cannulation significantly faster (median 20 s vs. 63 s). The overall complication rates were comparable in both groups. The limitations of this study are the relatively small number of patients and the slightly unequal number of patients in the study arm [14]. Hausmann et al. conducted a multicenter randomized trial to compare the performance of four different modern guidewires in different ERCP scenarios, analyzing tip shape (angled vs. straight). The primary outcome was the success rate of ERCP completion, while procedure time was a secondary endpoint. The results showed that angled-tip guidewires had the highest success rate for native papilla intubation (88.9%) and overall interventions (87.5% vs. 79.9%), particularly for intrahepatic strictures (89.5% vs. 67.8%). Crossover attempts favored angled-tip guidewires, adding 12.6% success after straight-tip failure, while switching to straight-tip guidewires after angled-tip failure provided minimal benefit (3.4%). Based on these findings, angled-tip guidewires can be recommended as the first choice for ERCP, especially for intrahepatic strictures. The limitations are the non-blinded design of the study, the only use of hydrophilic guidewires, and the relatively small number of patients included in the pancreatic group [15]. Pancreatic duct cannulation is often challenging due to anatomical factors. While traditional J-tipped guidewires are useful for biliary cannulation [16], their large tip radius makes them unsuitable for pancreatic interventions. However, a newly developed small J-tipped guidewire with a reduced loop size has enabled safer and more effective pancreatic procedures. A retrospective single-center study comparing this guidewire to an angled-tip guidewire in 294 patients (114 in the small J group and 180 in the angled group) found a significantly higher success rate with the small J-tipped guidewire (76% vs. 47%, *p* < 0.001). It remained an independent predictor of success, even in complex cases like chronic pancreatitis or sharp ductal angles. Additionally, in a retrospective study, the small J-tipped guidewire was associated with a lower incidence of post-ERCP pancreatitis (3.5% vs. 12.2%, *p* = 0.01), likely due to its looped tip reducing pancreatic duct trauma [17] (Figure 2).

### 2.5. Stiffness and Maneuvrability

The effectiveness of guidewires in endoscopic procedures depends on the balance between friction and flexibility. Hydrophilic, flexible-tip guidewires facilitate selective ductal access but may lack sufficient friction for countertraction, hindering device advancement. In contrast, stiff, large-caliber guidewires enhance device advancement by minimizing lateral deviation and transmitting forward axial forces. Navigating narrow or irregular spaces often requires soft, small-caliber, angled, or highly flexible guidewires [12]. After cannulation with a hydrophilic guidewire, as for small-diameter wires, it is suggested to exchange it for a more stable one to maintain position and improve stability during stent or device introduction. A recent structural study on eight different guidewires analyzed how core wire diameter, coating type, and tip characteristics influence clinical performance. The study found that shaft stiffness depends primarily on core diameter rather than coating thickness. Therefore, to increase shaft stiffness while maintaining the same material and specifications, a thicker core wire with a thinner surface coating is preferable. Similarly, a larger core enhances torque response by increasing torsional stiffness. Among guidewires with identical cores, those with uneven coatings like Fielder25 (Asahi Intecc, Tokyo, Japan) or EndoSelector (Boston Scientific, USA) demonstrate significantly lower frictional resistance and increased shaft lubricity, leading to a better torque response rate compared to flat-coated guidewires like VisiGlide 2 (Olympus, Japan) and M-Through (Asahi Intecc, Tokyo, Japan). In conclusion, the seeking ability is influenced by the torque response and tip core weight. To optimize performance, modern guidewires combine a hyperflexible, hydrophilic tip for navigating strictures with a stiff shaft to facilitate device passage [18]. An RCT by Jörg et al. found that VisiGlide (Olympus, Japan), a hyperflexible hydrophilic tip with a stiff shaft, improved stricture cannulation success and reduced procedure time compared to Radifocus (Terumo, Japan), a flexible hydrophilic wire. The limitation of this study is that it did not differentiate between the impact of the guidewire on the successful cannulation of the papilla versus negotiating the biliary stricture, and instead, it calculated only the combined success rate; additionally, the study’s findings cannot be obviously generalized to other untested guidewires [19]. Another RCT by Park et al. showed that VisiGlide 2 (Olympus, Japan), with its highly flexible tip, facilitated selective biliary cannulation, reducing cannulation time and papillary attempts compared to the conventional Jagwire (Boston Scientific, USA). The limitations of the study are the difference in diameter between the two guidewires, the small amount of patients enrolled, and the single-center design [20].

## 3. ERCP

ERCP is a well-established endoscopic technique widely used in clinical practice for the treatment of biliopancreatic diseases performed with the aid of a duodenoscope, which allows for access to the second portion of the duodenum and, through the papilla of Vater, to the biliary and pancreatic systems. To support clinical decision-making, Figure 3 illustrates a stepwise algorithm for guidewire selection in ERCP, stratified by indication and designed to adapt to cannulation failure or anatomical challenges.

### 3.1. Biliary Duct Cannulation

Guidewires are essential in the early stages of ERCP. A pivotal 2004 study by Fausto Lella demonstrated the superiority of wire-guided cannulation (WGC) over contrast-assisted methods [21]. The WGC technique has widely become the first-line approach to biliary cannulation [22]. Both techniques require fluoroscopy, as guidewires, like contrast agents, have a radiopaque core visible on radiological images. In WGC cannulation, the guidewire is introduced through the papilla using a sphincterotome or cannulotome, and fluoroscopy enables real-time visualization for selective biliary or pancreatic duct cannulation (Figure 4a). Studies show that this technique increases the success rate, reduces papillary trauma, and minimizes contrast injection into the Wirsung duct, lowering post-ERCP pancreatitis rates [23]. A meta-analysis of RCTs confirmed that WGC cannulation significantly improves biliary cannulation rates (OR: 2.05, CI: 1.27–3.31) and decreases post-ERCP pancreatitis (OR = 0.23, CI: 0.13–0.41) [24]. In the case of trainees, who often take longer to perform ERCP, an RCT indicated that angled guidewires led to higher successful selective biliary cannulation rates compared to straight guidewires. The limitations of the study are the small sample size and the definition of difficult biliary cannulation set at 7 min, which could be equivocal [25]. The typical WGC involves loading a straight guidewire onto a sphincterotome, but for trainees with limited experience, the angle of an angled one can assist in achieving successful cannulation. A recent RCT involving 588 patients compared exclusive WGC with the hybrid technique, in which small amounts of contrast are injected into the distal bile and pancreatic ducts when the infundibulum of the papilla of Vater is cannulated to help orient the papillotome or guidewire axis toward the biliary tree. The results of this study showed faster biliary cannulation, a shorter total ERCP time, reduced need for precut, and less inadvertent intubation of the Wirsung duct, which was identified as a greater risk factor for PEP than inadvertent contrast injection into the pancreatic duct (OR 4.22 vs. 2.63). However, along with these potential beneficial factors, PEP occurred at similar rates in both groups. The limitations of the study include its single-center design, the lack of a standardized threshold for precutting, and the absence of prophylactic measures in both groups [26]. Selective biliary cannulation using conventional techniques fails in up to 18% of native papilla cases [22]. In difficult cases, the “double guidewire technique” (DGW-T) is often employed, where one wire is placed in the Wirsung duct to block the pancreatic orifice, aiding the second wire in cannulating the bile duct (Figure 4b). This technique also helps in identifying papillary anatomy and orienting biliary access [27].

### 3.2. Pancreatic Duct Cannulation

Selective pancreatic duct cannulation is indicated for various pancreatic disorders, such as idiopathic recurrent acute pancreatitis, chronic pancreatitis, pancreatic ductal injuries, fistula formation, and sphincter of Oddi dysfunction (Figure 5) [22,28]. Pancreatic duct access can be challenging either due to the smaller caliber of the duct or the presence of pathological conditions such as stenosis, stones, or altered anatomy. Typically, smaller-caliber guidewires (0.025- or 0.018-inch) and fully hydrophilic guidewires are used. In very difficult cases, an advanced technique, known as either the reverse double-guidewire cannulation or the bile duct guidewire indwelling method according to different authors, has been described. In this technique, the first guidewire is placed into the biliary duct, which helps keep the papilla open and stabilized, pulls the septum superiorly, and straightens the pancreatic duct, facilitating its cannulation with a second guidewire. This approach can also be useful in cases of parapapillary diverticulum with a deviated papilla and increased papillary mobility, as well as in patients with a tortuous and elongated papillary sphincter muscle [29]. Notably, a case report describes the successful use of an uneven double-lumen cannula (0.025- and 0.035-inch lumens) for the same purpose [30]. Access to the main pancreatic duct (MPD) through the major papilla is sometimes impossible due to pancreas divisum, a distortion of Wirsung’s duct, or other anatomical variations, necessitating minor papilla cannulation [31]. In this setting, experts recommend WGC using small-caliber guidewires (0.018- and 0.021-inch), with or without contrast injection, followed by a standard pull-type sphincterotomy or a needle-knife sphincterotomy over a stent [22]. When the deep cannulation of the dorsal duct with a pull-sphincterotome fails, wire-assisted access sphincterotomy is an alternative technique. After successful deep cannulation with a guidewire, the sphincterotome is exchanged for a needle-knife, which is advanced alongside the wire; then, the minor papilla is incised using the wire as an anatomic guide. Compared to freehand precut sphincterotomy, it ensures early and stable wire access. Additionally, unlike needle-knife sphincterotomy over a stent, it allows for further interventions without requiring stent removal [32].

## 4. EUS

EUS-guided procedures, including EUS-guided biliary drainage (EUS-BD), pancreatic drainage (EUS-PD), and drainage of pancreatic fluid collections, are increasingly being used in clinical practice [33]. A key advantage of EUS guidance is its flexibility in choosing access points based on patient and ductal anatomy. In this context, selecting the optimal puncture site and effectively manipulating the guidewire within the ducts are crucial steps. Figure 6 provides a procedural algorithm outlining primary and secondary guidewire choices, highlighting contingency strategies in case of a failed wire advancement. Notably, a 19-gauge EUS needle can accommodate guidewires up to 0.035 inches, while a 22-gauge needle is limited to guidewires up to 0.021 inches [2].

### 4.1. EUS-Guided Biliary Drainage (EUS-BD)

When ERCP fails or is not feasible (e.g., due to papillary tumor infiltration or surgically altered anatomy), EUS-guided biliary drainage (EUS-BD) represents a viable alternative. The main approaches include EUS–hepaticogastrostomy (EUS-HGS), rendezvous (EUS-Rv), and antegrade stenting (EUS-AG), each requiring specific guidewire strategies [34,35]. EUS-HGS typically involves puncturing the left intrahepatic bile duct from the stomach under EUS guidance (Figure 7a), followed by guidewire advancement, tract dilation, and stent placement (Figure 7b). A 19-gauge needle with a 0.025- or 0.035-inch guidewire is generally preferred for its stiffness and maneuverability [36], although a 22-gauge needle with a 0.018-inch wire may improve access to non-dilated ducts [37,38]. Devices compatible with 0.018-inch wires are now available [39,40], and in select cases, tract dilation may be avoided to reduce bile leakage risk [41]. Guidewire manipulation is often the most challenging step [42,43], particularly when an acute needle–duct angle (<85°) limits advancement [43], while obtuse angles (>135°) are associated with better outcomes [44]. Several rescue techniques have been described for failed guidewire passage, including exchanging the wire, liver impaction of the needle, use of an uneven catheter or balloon-assisted methods, and re-puncture of another duct [45]. Techniques such as the “moving-scope” or “jumping” methods have been reported to facilitate wire passage in difficult cases [46,47].

EUS-Rv and EUS-AG also rely on effective guidewire manipulation. In EUS-Rv, the guidewire is advanced in an antegrade manner into the duodenum or jejunum and retrieved endoscopically to complete biliary access: a long guidewire is ideal to facilitate scope exchange [2]. Failure in the rendezvous technique often occurs when the wire cannot pass through a stricture or the major papilla: a 0.018-inch guidewire may improve success in these cases [48]. In EUS-AG, particularly for stone extraction, a stiff 0.035-inch wire can help straighten the tract and ease antegrade advancement of catheters, especially when dealing with unfavorable angulation at the hepatic hilum or papilla [49].

### 4.2. EUS–Pancreatic Drainage (EUS-PD)

Two main approaches for EUS-PD exist: EUS-Rv, where a small-caliber wire is advanced in an antegrade manner into the duodenum for ERCP stenting (Figure 8), and EUS-PD transmural drainage, involving serial dilation and stent placement from the stomach or the duodenum (pancreaticogastrostomy/pancreaticoduodenostomy) [50]. Successful EUS-PD relies on precise pancreatic duct puncture and effective guidewire manipulation, given the small duct size, side branches, and potential strictures. A 19-gauge needle with a sharp tip and a 0.025-inch flexible-tip guidewire is generally recommended [51]. In cases with a small pancreatic duct, a 22-gauge needle combined with a smaller (0.018-or 0.021-inch) guidewire may be used, though these thinner guidewires can be challenging to manipulate even after successful duct puncture [52]. The advantage of the 22-gauge needle lies in its ability to puncture even a fibrotic pancreas or a stiff, narrow MPD; in their single-center retrospective study, Matsunami et al. reported high technical success in EUS-PD using a 22-gauge needle despite targeting a small MPD (median diameter 3.5 mm, range 1–14 mm) [53]. However, due to the limited fluoroscopic visibility and insufficient shaft stiffness for therapeutic device support that 0.018- and 0.021-inch guidewires typically offer, when initial access is achieved with a smaller guidewire, a subsequent exchange to a larger-caliber (0.025- or 0.035-inch), stiffer wire is often necessary to ensure procedural success [54]. Studies such as those by Inoue et al. [55] and Sasaki et al. [56] have noted the value of an ultra-tapered or hybrid wire with a stiff body to facilitate pushability and a soft tip to minimize ductal trauma. For patients with surgically altered anatomy, such as those who have undergone gastrectomy, selecting an appropriate puncture site can be difficult due to the reduced stomach volume. In such cases, transjejunal EUS-PD is an ideal approach, as the jejunum and pancreatic duct are closely anastomosed. A stiff guidewire, typically used for endoscopic long intestinal tube insertion, helps optimize this technique. After straightening the twisted afferent jejunal loop using an enteroscope, the guidewire is placed to maintain the loop’s shape. Additionally, by deploying the guidewire into the working channel of a forward-viewing echoendoscope (ropeway method), smooth scope insertion is achieved, facilitating the transjejunal EUS-PD procedure [57].

### 4.3. Other EUS-Guided Procedures

EUS-guided procedures have increasingly adopted lumen-apposing metal stents (LAMSs), revolutionizing various interventions by simplifying access and enhancing procedural efficiency. One of the most significant applications is in the drainage of pancreatic fluid collections or walled-off necrosis, where traditional transmural drainage relied on double-pigtail plastic stents and long guidewires to facilitate device exchanges [2]. The introduction of electrocautery-enhanced LAMSs has minimized the need for guidewires in freehand techniques; however, they remain essential after initial stent deployment. Guidewires can be preloaded into the stent catheter or inserted post-placement to aid further interventions, facilitate LAMS dilation, and enable the coaxial placement of plastic stents [2]. In cases where echoendoscope instability is a concern, placing a guidewire before LAMS deployment can improve stability, reduce technical complications, and serve as a rescue option, such as the “LAMS-in-LAMS” technique [58]. In EUS-guided gastroenterostomy (GEA), LAMSs allow for both direct puncture and guidewire-assisted techniques, though the latter may increase the risk of stent misdeployment [59], due to the potential for guidewire manipulation to displace the bowel [60]. Similarly, EUS-guided choledochoduodenostomy (EUS-CDS) and gallbladder drainage (EUS-GBD) have become effective alternatives for managing biliary obstructions and acute cholecystitis (Figure 9) [61]. LAMSs have significantly improved these procedures, enabling both freehand and over-the-guidewire techniques. Typically, a 19-gauge needle is used for puncture, with guidewires of 0.035-inch or 0.025-inch diameter facilitating stent placement. The 0.025-inch Jagwire Revolution (Boston Scientific, USA) has been noted for its enhanced stiffness, providing a balance between rigidity and maneuverability, which fits for this kind of procedures [34]. In EUS-GBD, the use of a thin 22-gauge EUS-FNA needle can reduce complications like bile leakage, while the 0.018-inch guidewire (e.g., Fielder 18, Asahi Intecc, Japan) has shown promise in improving procedural outcomes [62]. Overall, LAMSs have streamlined EUS-guided procedures, reducing the need for repeated guidewire exchanges while maintaining their role in ensuring technical success.

## 5. Guidewire-Related Complications and Safety

### 5.1. ERCP-Related Complications

Guidewires are generally safe and easy to handle, with the most significant directly related issue being bile or pancreatic duct perforation (type 3 perforation following Stapfer classification), which is rare and typically identified during the procedure by contrast leakage outside the biliary or pancreatic system [63]. Guidewire-related perforation often occurs locally around the ampulla of Vater forcing the entry of a guidewire when biliary cannulation is difficult [64]. A meta-analysis found that ERCP-related perforations remain uncommon (0.39%), of which guidewire-related perforations accounted for 16% [65]. Management usually involves conservative treatment, including antibiotic therapy and the placement of a fully covered self-expandable metal stent (FC-SEMS) if the bile duct is ultimately cannulated [66,67]. Reported mechanism of fractured guidewire include forced traction, imperfection in the painted Teflon guidewire coating, electrical short circuits between cutting wire and guidewire, and detachment of the floppy tip from the shaft. Successful retrievals of the fractured guidewire from the MPD are rarely reported [68]. Acute and long-term biliopancreatic complications are only anecdotally reported, such as a case of cholangitis related to hydrophilic guidewire fracture [69]. Therefore, a conservative management approach can be suggested without the need to retrieve the wire fragment [68]. Inadvertent portal vein cannulation or stenting has also been reported in the literature, especially in cases of malignant etiologies that distort local anatomy. While the majority of portal vein cannulation or stenting merely needs the withdrawal of a catheter or stent, an interventional radiologist should be involved early, given an unexpected risk of torrential bleeding that may require portal vein or common bile duct stent grafting [70].

### 5.2. EUS-Related Complications

In performing therapeutic EUS, guidewire shearing is a potential complication in which the coating of the guidewire can be stripped off due to contact with the sharp beveled needle tip, leaving a foreign body in the pancreaticobiliary tract that may act as a nidus for infection. Shearing is most likely to happen when the guidewire is pulled back into the needle after having reached the periphery of the duct. This can be prevented by slightly withdrawing the needle while maintaining access [71]. To avoid guidewire shearing, careful manipulation is essential. The guidewire should not be repeatedly pushed and pulled, as this increases the risk of damage. Additionally, minimizing the angle between the needle and the duct, using a flexible-tip wire, or employing a coaxial microcatheter can help prevent such incidents. While a PTFE press coating can help prevent guidewire shearing, it also increases friction resistance compared to jacket-type guidewires. This resistance can make guidewire manipulation and device exchange challenging, especially in the presence of bile or a contrast medium. To address this limitation, a novel hybrid guidewire (CAPELLA; Japan Lifeline Co., Ltd., Tokyo, Japan) has recently been introduced. This guidewire features a PTFE press coating from the tip to 195 mm, while the remaining portion is a grooved jacket-type design. This dual structure offers two key advantages: reducing the risk of shearing while minimizing friction resistance [72].

## 6. Conclusions

Guidewire selection in ERCP and EUS-guided drainage plays a crucial role in optimizing technical success while minimizing complications. In ERCP, reducing contrast injections and repeated pancreatic duct cannulations remains essential to lowering the risk of PEP [73]. Studies highlight the importance of selecting an appropriate guidewire to facilitate deep biliary access while avoiding inadvertent pancreatic manipulation. Hydrophilic guidewires with angled tips have demonstrated superior performance in selective biliary cannulation and intrahepatic stricture management [14,15], while small J-tipped guidewires offer advantages in pancreatic duct interventions by improving success rates and reducing the risk of PEP [17]. In the context of EUS-BD, factors such as the angle of needle insertion and stable guidewire positioning are critical in preventing complications like shearing or fracture. A growing body of literature highlights the increasing adoption of 0.018-inch guidewires paired with specialized microcatheters to enhance procedural efficiency [37,48,62]. Additionally, the emergence of LAMS with integrated electrocautery capabilities is shifting procedural paradigms by potentially reducing or even eliminating the need for prolonged guidewire use [61]. Current evidence underscores that a strategic approach to guidewire selection, combined with prophylactic interventions, can significantly improve procedural outcomes and reduce morbidity in advanced biliopancreatic endoscopy. Despite these innovations, no single guidewire is universally superior, emphasizing the need for tailored selection based on patient-specific factors and procedural goals. Future research should focus on optimizing guidewire characteristics, improving ease of use, and reducing complications to further enhance outcomes in biliopancreatic endoscopy.

## Figures and Tables

**Figure 1 medicina-61-00913-f001:**
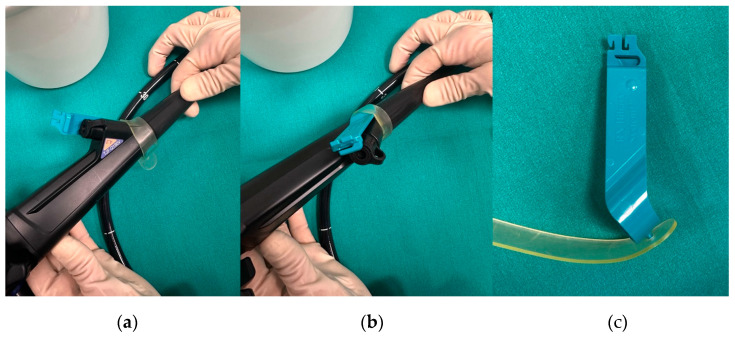
(**a**,**b**) Two views from different angles of the RX locking system (Boston Scientific) mounted on the duodenoscope; (**c**) the RX locking device disassembled, showing its components: a rubber clamp that is fixed externally to the working channel port and a plastic locking mechanism that secures the guidewire in place.

**Figure 2 medicina-61-00913-f002:**
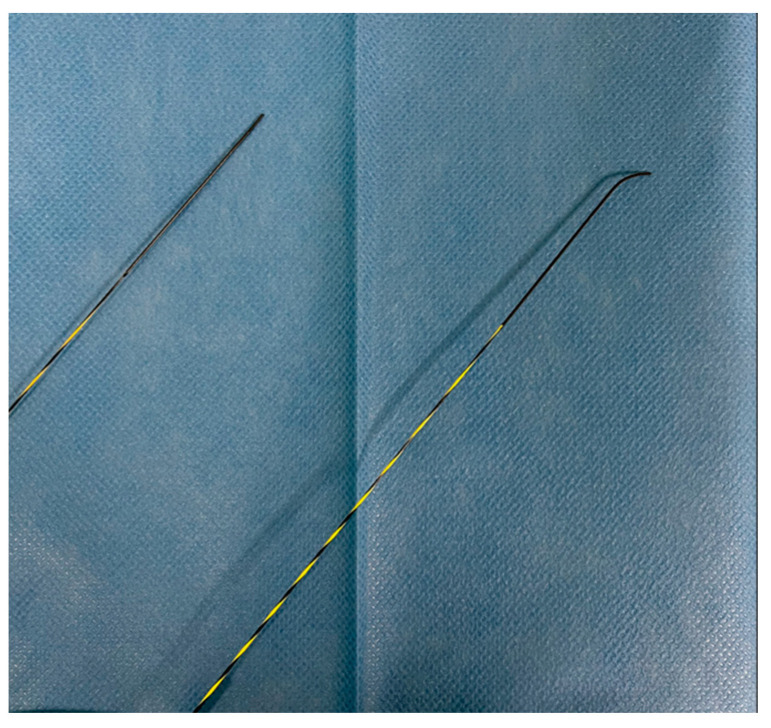
Comparison between straight-tip and angled-tip guidewires. The angled tip facilitates selective cannulation and navigation through strictures or tortuous anatomy, while the straight tip may offer enhanced pushability in straight trajectories. These configurations are chosen based on procedural needs and anatomical challenges.

**Figure 3 medicina-61-00913-f003:**
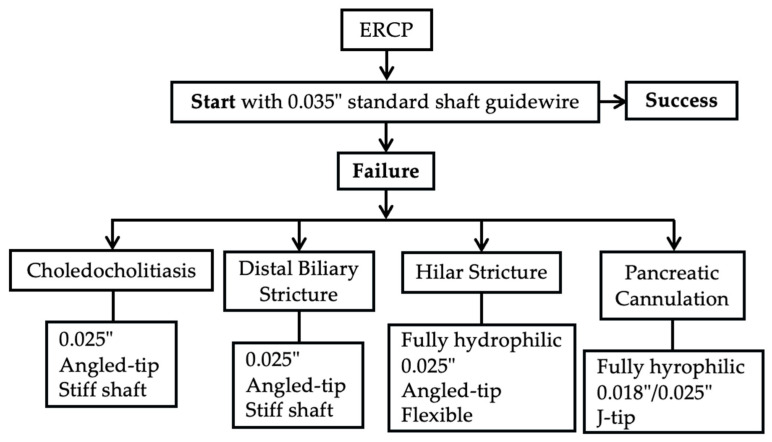
Algorithm for guidewire selection to optimize ERCP success. The procedure typically begins with a 0.035” hydrophilic-tipped, standard shaft guidewire. In case of technical failure, guidewire choice is adapted based on the clinical indication such as choledocholithiasis, distal or hilar strictures, or pancreatic duct cannulation, with tailored strategies for each scenario to enhance cannulation success and procedural efficiency.

**Figure 4 medicina-61-00913-f004:**
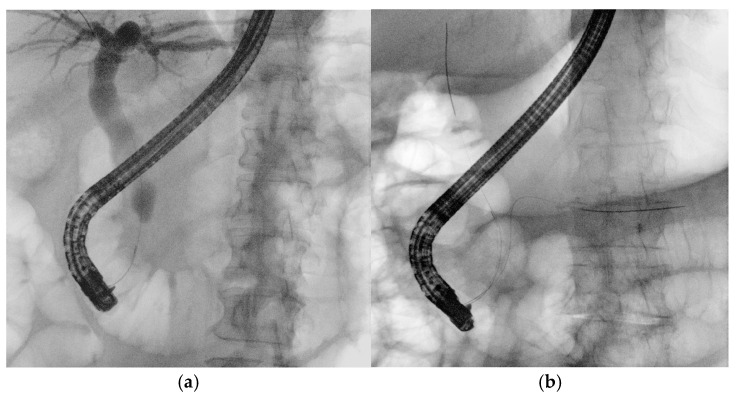
(**a**) Wire-guided cannulation in ERCP; (**b**) double-guidewire technique: the first guidewire is positioned in the main pancreatic duct, followed by placement of the second guidewire into the biliary duct.

**Figure 5 medicina-61-00913-f005:**
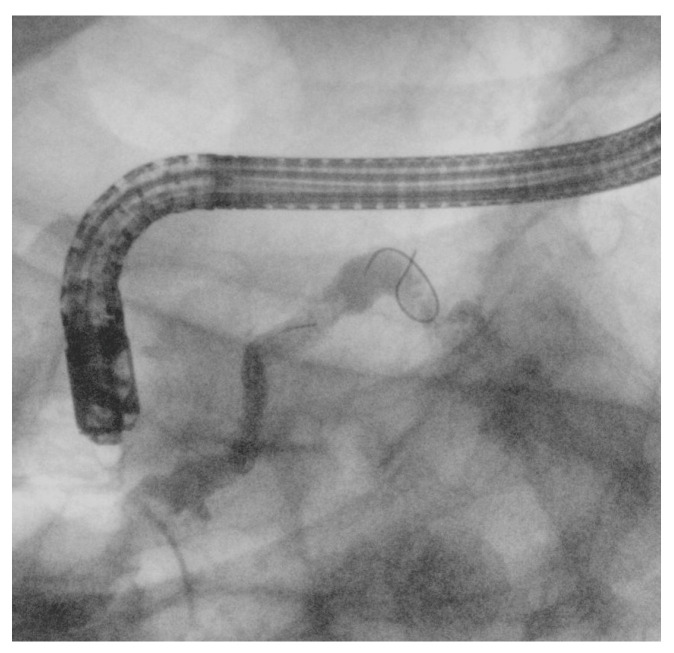
Guidewire-assisted pancreatic cannulation followed by contrast injection showing a dilated and tortuous main pancreatic duct in the context of chronic pancreatitis.

**Figure 6 medicina-61-00913-f006:**
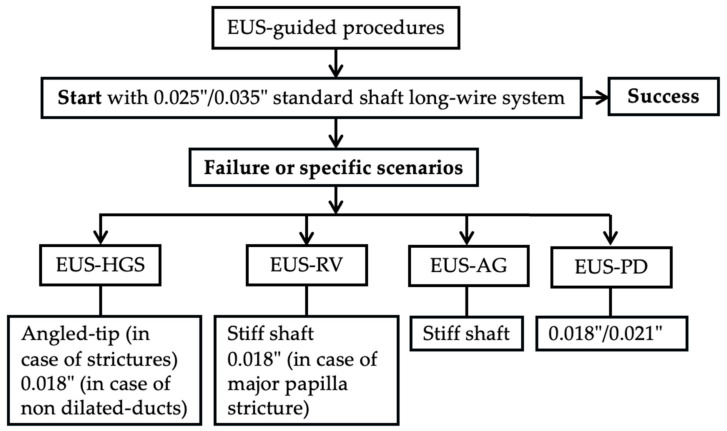
Algorithm for guidewire selection in EUS-guided procedures. Initiation with a 0.025”/0.035” hydrophilic-tipped guidewire is generally recommended, with adjustments based on failure or specific procedural contexts such as EUS-HGS, EUS-RV, EUS-AG, and EUS-PD, highlighting tailored guidewire characteristics suited to each approach.

**Figure 7 medicina-61-00913-f007:**
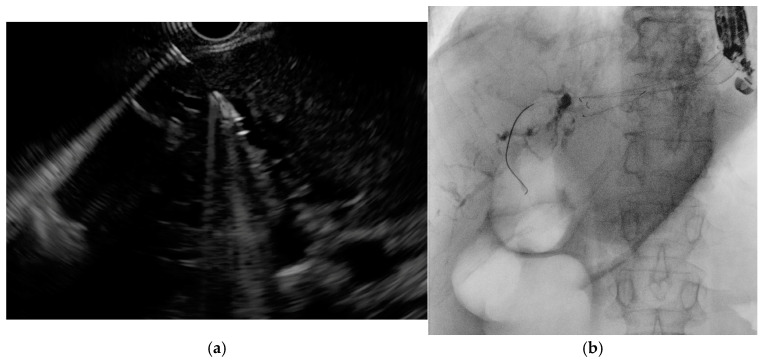
(**a**) EUS-guided puncture of intrahepatic bile duct with subsequent guidewire advancement; (**b**) placement of a self-expandable metal stent across the hepaticogastrostomy secured over the guidewire.

**Figure 8 medicina-61-00913-f008:**
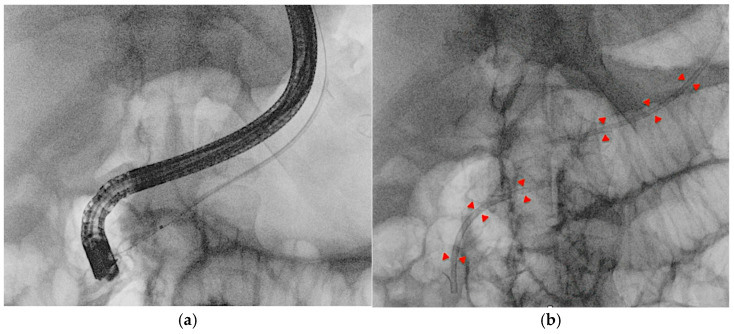
Radiological image of the EUS-PD rendezvous technique. (**a**) After placing a stent across pancreaticogastrostomy, the guidewire is advanced in an antegrade manner into the pancreatic duct under EUS guidance, followed by standard ERCP and transpapillary stent placement over the wire for pancreatic duct drainage. (**b**) The end of the procedure with the two stents (highlighted by the red arrows) in the main pancreatic duct.

**Figure 9 medicina-61-00913-f009:**
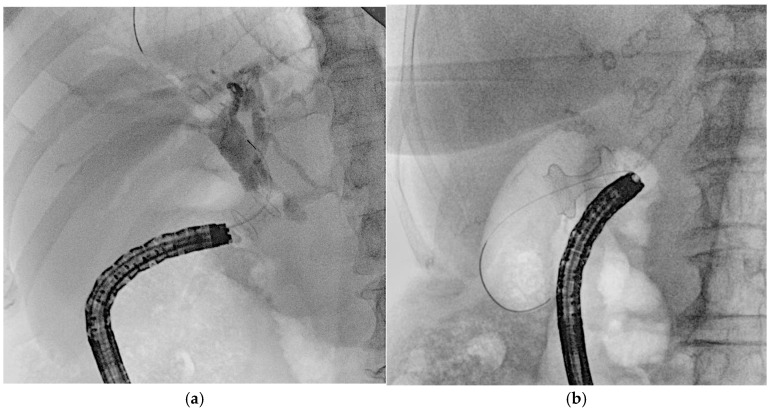
(**a**) Guidewire-assisted advancement of a double-pigtail stent through an 8 × 8 mm lumen-apposing metal stent (LAMS) used for choledochoduodenostomy (EUS-CDS) to secure biliary drainage and prevent LAMS obstruction; (**b**) EUS-guided gallbladder drainage (EUS-GBD) performed using an electrocautery-enhanced lumen-apposing metal stent (LAMS). A 0.035-inch guidewire was used to improve stability and maintain access during deployment.

**Table 1 medicina-61-00913-t001:** Physical characteristics of guidewires for biliopancreatic endoscopy.

Structure	Length	Diameter	Tip Shape	Stiffness
Hydrophobic shaft with hydrophilic tip	Long wire system (450, 480 cm)	Standard (0.035-inch)	Straight	Standard
Fully hydrophilic	Short wire system (260 cm)	Thin (0.018- and 0.025-inch)	Angled, “J”-shaped	Stiff, ultra stiff

**Table 2 medicina-61-00913-t002:** Technical specifications of the most common available guidewires for biliopancreatic endoscopy.

Guidewire	Manufacturer, Country	Diameter (inch)	Length (cm)	Shaft Style	Tip Shape	Tip (cm)
Fielder25	Asahi, Intecc, Tokyo, Japan	0.025	450	Standard (flexible)	Straight, angled	7.5
M-Through	Asahi, Intecc, Tokyo, Japan	0.025	450	Standard (stiff), Flex	Straight, angled	7.5
Dreamwire	BS, Marlborough, MA, USA	0.035	260, 450	Standard, stiff	Straight, angled	10
EndoSelector	BS, Marlborough, MA, USA	0.025	450	Standard	Staight, angled	7
Hydra Jagwire	BS, Marlborough, MA, USA	0.035	260, 450	Standard, stiff	Straight, angled	5, 10
Jagwire	BS, Marlborough, MA, USA	0.025, 0.035	260, 450	Standard, stiff	Straight, angled	5
Jagwire Revolution	BS, Marlborough, MA, USA	0.025	260, 450	Standard	Straight, angled	5
NaviPro	BS, Marlborough, MA, USA	0.018, 0.025, 0.035	260	FH Standard, stiff	Straight, angled	3
NovaGold	BS, Marlborough, MA, USA	0.018	260, 480	Standard	Straight	6
XWire	ConMed, Westborough, MA, USA	0.025, 0.035	260, 450	Standard, stiff	Straight, angled	5
FXWire	ConMed, Westborough, MA, USA	0.035	260, 450	Standard	Straight, angled	5
Acrobat 2	Cook Medical, Bloomington, IN, USA	0.025, 0.035	260, 450	Standard	Straight, angled	4
Tracer Metro	Cook Medical, Bloomington, IN, USA	0.021, 0.025, 0.035	260, 480	Standard	Straight, angled	5
Roadrunner	Cook Medical, Bloomington, IN, USA	0.018	260, 480	Standard	Straight, angled	3
Delta Wire	Cook Medical, Bloomington, IN, USA	0.025, 0.035	260	FH	Straight, angled	N/A
GPS Wire	Fujifilm, Tokyo, Japan	0.035	260, 450	Standard	Straight, J-tip	N/A
J-Wire Series	J-MIT, Tokyo, Japan	0.018, 0.025	450	N/A	Angled	5
RewoWave	Olympus, Tokyo, Japan	0.025, 0.035	260, 450	Standard, stiff, ultra stiff	Straight, angled	5, 7
VisiGlide, VisiGlide 2	Olympus, Tokyo, Japan	0.025, 0.035	270, 450	Standard	Straight, angled	7
Glidewire	Olympus, Tokyo, Japan	0.018, 0.025, 0.035	260, 450	FH Standard, stiff	Straight, angled	3, 5, 8
Optimos	Taewong, China	0.025, 0.035	260, 450	Standard	Straight, angled	5.5, 6, 7, 7.5
Radifocus	Terumo, Tokyo, Japan	0.018, 0.025, 0.035	260, 450	FH Standard, stiff	Straight, angled, J-tip	3

BS, Boston Scientific; FH, fully hydrophilic; N/A, not available.

## Data Availability

Not applicable.

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
