# Peer review of "The Role and Appropriate Selection of Guidewires in Biliopancreatic Endoscopy"

_medicina, 2025, doi:10.3390/medicina61050913_

Round 1

Reviewer 1 Report

Comments and Suggestions for Authors

The manuscript is an excellent review on the use of guidewires in endoscopic hepatobiliary and pancreatic therapy.

The most commonly available guidewires and their respective manufacturers are listed.

The two endoscopic techniques for biliopancreatic drainage are endoscopic retrograde cholangiopancreatography (ERCP) and endoscopic ultrasonography (EUS). Guidewires are used to access the bile and pancreatic ducts, afterwards different devices, such as catheters, plastic or self-expanding metallic stents, can

In my opinion, the review is adequate. In order to publish it, the three points pointed out to the authors should be corrected

 be passed over the guidewires to achieve drainage.

The authors' review is very detailed regarding the various ERCP and EUS techniques.

To improve the manuscript:

1) Regarding the disadvantages of long guidewires, perhaps mention should be made of post-ERCP pancreatitis due to poor communication between the endoscopist and assistant.

2) In Figure 2, a J-tip guidewire should be added to compare with straight and angled guidewires.

3) In Figure 4, it should be noted that regarding to the double guidewire technique, the guidewire are located in the gallbladder and pancreas.

4) Regarding the hepaticogastrostomy, it is stated that the puncture is performed from the stomach or duodenum. The duodenum must be removed.

5) In Figure 8 a), it appears that the choledochoduodenostomy was performed with LAMS other than the usual axios because there are marks at the ends and in the center.

6) Among the complications of the guidewires, when referring to type 3 perforation, it is mentioned that it can be resolved with a metal stent in the common bile duct. It should be added that "if the bile duct is ultimately cannulated."

Author Response

Comment 1: Regarding the disadvantages of long guidewires, perhaps mention should be made of post-ERCP pancreatitis due to poor communication between the endoscopist and assistant.

Response 1: Thank you for your insightful comment. We agree that post-ERCP pancreatitis is an important consideration, particularly when using long guidewires. In response, we have addressed this issue in the section on guidewire length, where we discuss the potential risks associated with poor communication between the endoscopist and assistant, which could contribute to complications such as post-ERCP pancreatitis. We hope this addition clarifies the risks associated with long guidewires in such procedures.

Comment 2: In Figure 2, a J-tip guidewire should be added to compare with straight and angled guidewires.

Response 2: Thank you for your suggestion. Unfortunately, we were unable to obtain an image of a J-tip guidewire for comparison with the straight and angled guidewires in Figure 2. However, we appreciate the importance of including this comparison and will consider including it in future revisions if we can access the necessary images.

Comment 3: In Figure 4, it should be noted that regarding to the double guidewire technique, the guidewire are located in the gallbladder and pancreas.

Response 3: Thank you for your helpful suggestion. We have made the requested change and modified Figure 4 to show the classic depiction of the double guidewire technique, with the guidewires placed in the common bile duct, rather than in the gallbladder. We believe this revision better aligns with standard clinical practice and enhances the clarity of the figure.

Comment 4: Regarding the hepaticogastrostomy, it is stated that the puncture is performed from the stomach or duodenum. The duodenum must be removed.

Response 4: Thanks, we removed that.

Comment 5: In Figure 8 a), it appears that the choledochoduodenostomy was performed with LAMS other than the usual axios because there are marks at the ends and in the center.

Response 5: Thank you for your observation. To avoid any potential confusion, we have replaced the ZEUS LAMS image in the figure with an image of the HOT AXIOS LAMS, which is more commonly used for choledochoduodenostomy procedures. We hope this change ensures clarity.

Comment 6: Among the complications of the guidewires, when referring to type 3 perforation, it is mentioned that it can be resolved with a metal stent in the common bile duct. It should be added that "if the bile duct is ultimately cannulated."

Response 6: Thanks, we added yuor specification in the text.

Reviewer 2 Report

Comments and Suggestions for Authors

Major 
1. The manuscript thoroughly reviews the technical characteristics of guidewires; however, it fails to translate this into actionable guidance for clinical decision-making. Despite providing extensive data, the review lacks a clear, concise summary—such as an algorithm or decision table—that helps the reader select a guidewire based on specific clinical scenarios (e.g., hilar stricture vs. distal bile duct stricture, EUS-HGS vs. EUS-Rv). This significantly limits the practical utility of the manuscript.

2. The review is unnecessarily verbose, with several sections repeating similar information. In particular, the sections discussing guidewire manipulation in EUS-HGS and EUS-Rv overlap in both structure and content. The authors should condense and reorganize the text to avoid redundancy and improve clarity. The current length may deter readership.

3. The manuscript is overly focused on standard ERCP and EUS procedures in adult patients, with no mention of guidewire selection in special or high-risk populations—e.g., pediatric patients, surgically altered anatomy (e.g., Roux-en-Y), or patients with complex strictures. This omission weakens the comprehensiveness of the review.

4. While the manuscript includes numerous citations, the authors often present study findings without sufficient critical appraisal. For instance, results from RCTs are presented at face value without discussing methodological limitations, heterogeneity, or risk of bias. A review article should not merely summarize but should also evaluate the strength of evidence.

Minor 
1. Several typographical errors (e.g., "lenght" instead of "length" in tables) and occasional awkward phrasing reduce the professionalism of the manuscript. A thorough proofreading and professional English editing are recommended before acceptance.

2.The figures are inadequately labeled and described. For example, Figure 2 comparing straight and angled tips lacks explanatory captions, making it difficult to interpret without cross-referencing the main text. Each figure should be self-contained and clearly indicate its relevance.

3. One of the authors is disclosed as a consultant for Boston Scientific and Olympus. Given that multiple guidewires from these companies are extensively discussed and favorably characterized, the manuscript should include a more detailed conflict of interest statement and explicitly clarify whether any commercial input influenced the manuscript content.

4. Despite its extensive content, the manuscript is missing a well-structured summary table comparing guidewires across parameters like flexibility, tip shape, torque response, and clinical application. This would enhance readability and usability for endoscopists.

Comments on the Quality of English Language

The manuscript is generally understandable and conveys the intended content; however, the quality of English requires moderate revision to meet the standards of a high-impact scientific journal. The following issues are noted:

Frequent typographical errors (e.g., “lenght” instead of “length”) appear throughout the text and tables, suggesting a lack of careful proofreading.

Several sentences are overly long or grammatically awkward, reducing clarity and readability. For example, paragraphs in sections 2.2 and 4.1.1 are verbose and would benefit from syntactic simplification.

Some technical terminology is inconsistently used, and the transition between sections could be smoother with more cohesive linking phrases.

A professional English language editing service is recommended to improve grammar, flow, and precision of terminology. While the scientific content is strong, linguistic weaknesses currently detract from its overall quality and readability.

Author Response

Comments and Suggestions for Authors

Major

Comment 1. The manuscript thoroughly reviews the technical characteristics of guidewires; however, it fails to translate this into actionable guidance for clinical decision-making. Despite providing extensive data, the review lacks a clear, concise summary—such as an algorithm or decision table—that helps the reader select a guidewire based on specific clinical scenarios (e.g., hilar stricture vs. distal bile duct stricture, EUS-HGS vs. EUS-Rv). This significantly limits the practical utility of the manuscript.

Response 1: Thank you for this insightful comment. In response, we have included two summary flowcharts specifically designed to enhance the clinical applicability of the manuscript. These visual tools outline guidewire selection strategies based on procedural context, including ERCP (e.g., hilar vs. distal strictures) and EUS-guided interventions (e.g., EUS-HGS, EUS-Rendezvous, EUS-PD). Our aim was to provide clear, scenario-based guidance to support endoscopists in their decision-making process. We hope these additions improve the practical utility of the review, as suggested.

Comment 2: The review is unnecessarily verbose, with several sections repeating similar information. In particular, the sections discussing guidewire manipulation in EUS-HGS and EUS-Rv overlap in both structure and content. The authors should condense and reorganize the text to avoid redundancy and improve clarity. The current length may deter readership.

Response 2: Thanks for your advice. As you suggested, we tried to condense and reorganize the entire manuscript, in particular the section discussing EUS-HGS and EUS-Rv.

Comment 3: The manuscript is overly focused on standard ERCP and EUS procedures in adult patients, with no mention of guidewire selection in special or high-risk populations—e.g., pediatric patients, surgically altered anatomy (e.g., Roux-en-Y), or patients with complex strictures. This omission weakens the comprehensiveness of the review.

Response 3: Thank you for your comment. While our clinical practice does not currently involve pediatric patients, we have made an effort to address the issue of special populations in our review. Regarding pediatric patients, we were unable to find solid data or specific recommendations on the preference for certain guidewires in this population. As such, we have focused on surgically altered anatomy, such as Roux-en-Y gastric bypass, in the section on guidewire length.

Comment 4: While the manuscript includes numerous citations, the authors often present study findings without sufficient critical appraisal. For instance, results from RCTs are presented at face value without discussing methodological limitations, heterogeneity, or risk of bias. A review article should not merely summarize but should also evaluate the strength of evidence.

Response 4: Thank you for the clarification. We appreciate your suggestion to provide a more critical appraisal of the studies cited. In response, we have revisited the manuscript to include a discussion of the methodological limitations, potential biases, and heterogeneity of the studies, particularly the randomized controlled trials (RCTs) referenced. We have made sure to evaluate the strength of the evidence presented in the review, highlighting where limitations may affect the interpretation of results. We hope this enhances the clarity and rigor of our analysis.

Minor 

Comment 1: Several typographical errors (e.g., "lenght" instead of "length" in tables) and occasional awkward phrasing reduce the professionalism of the manuscript. A thorough proofreading and professional English editing are recommended before acceptance.

Response 1: Thank you for pointing this out. We have carefully revised the manuscript and all tables to correct typographical errors, including the noted misspelling of “length.” In addition, we have performed thorough proofreading and English editing throughout the text to improve clarity, consistency, and overall readability. We appreciate your attention to detail and your helpful suggestion.

Comment 2: The figures are inadequately labeled and described. For example, Figure 2 comparing straight and angled tips lacks explanatory captions, making it difficult to interpret without cross-referencing the main text. Each figure should be self-contained and clearly indicate its relevance.

Response 2: Thank you, we appreciate your input in helping us improve the clarity and utility of the visual materials. We have revised the figures to ensure they are more clearly labeled and self-explanatory. Specifically, we have updated the caption for Figure 2 to include a description of the differences between straight and angled tips and clarified its clinical relevance.

Comment 3: One of the authors is disclosed as a consultant for Boston Scientific and Olympus. Given that multiple guidewires from these companies are extensively discussed and favorably characterized, the manuscript should include a more detailed conflict of interest statement and explicitly clarify whether any commercial input influenced the manuscript content.

Response 3: Thank you for your observation. The disclosed conflict of interest refers to broadly relevant industry ties, but has no bearing on the present review, which received no commercial input. Mentions of guidewires from Boston Scientific and Olympus are solely based on their prominence in the literature, with no intent to promote any specific brand. Nonetheless, we clarified this further in the conflict of interest statement.

Comment 4: Despite its extensive content, the manuscript is missing a well-structured summary table comparing guidewires across parameters like flexibility, tip shape, torque response, and clinical application. This would enhance readability and usability for endoscopists.

Response 4: Thank you for your observation. We included a comparative table highlighting key structural parameters (length, diameter, and shaft design) consistently reported by manufacturers and relevant to procedural choices. While torque response is clinically important, it correlates with stiffness and is not independently disclosed or standardized, making it unsuitable for tabulation. Guidewire selection was addressed based on functional characteristics rather than brand names, and we added flowcharts to support clinical decision-making in ERCP and EUS, as suggested.

Comments on the Quality of English Language

The manuscript is generally understandable and conveys the intended content; however, the quality of English requires moderate revision to meet the standards of a high-impact scientific journal. The following issues are noted:

Frequent typographical errors (e.g., “lenght” instead of “length”) appear throughout the text and tables, suggesting a lack of careful proofreading.

Several sentences are overly long or grammatically awkward, reducing clarity and readability. For example, paragraphs in sections 2.2 and 4.1.1 are verbose and would benefit from syntactic simplification.

Some technical terminology is inconsistently used, and the transition between sections could be smoother with more cohesive linking phrases.

A professional English language editing service is recommended to improve grammar, flow, and precision of terminology. While the scientific content is strong, linguistic weaknesses currently detract from its overall quality and readability.

Response to “Comments on the Quality of English Language” : We thank the reviewer for this careful assessment of our manuscript’s language. In response, we have thoroughly re-read and revised the text, correcting all typographical errors, shortening and simplifying overly long or awkward sentences, and ensuring consistent use of technical terminology.

Round 2

Reviewer 2 Report

Comments and Suggestions for Authors

All major and minor reviewer comments were addressed in a meaningful way. The revised manuscript now offers clearer clinical guidance, improved structure, and more rigorous evaluation of evidence. Minor English polishing may still be beneficial.